# A Systematic Literature Review on Migration and Remittances in Mountainous Regions: Key Takeaways for Phuthaditjhaba, Free State, South Africa

Regret Sunge * and Calvin Mudzingiri

Department of Economics, Faculty of Economics and Management Sciences, University of Free State, Phuthaditjhaba 9866, Free State, South Africa; mudzingiric@ufs.ac.za
* Correspondence: sunger1@ufs.ac.za

**Abstract:** Remittances are essential to the sustainability of economies in mountainous regions that face massive labour migration due to limited income generation and employment opportunities. In 2021, the share of remittances in GDP in the top 10 mountainous economies in the world was over 20%. Nonetheless, most are characterised by relatively lower GDP per capita and high poverty levels. Drawing a comparison with other mountainous areas, Phuthaditjhaba, an emerging mountainous city of South Africa on the border with Lesotho, faces similar out-migration and inferior socio-economic parameters. A global systematic literature review on the impact of remittances on livelihoods, specifically targeting mountainous areas, is missing. We, therefore, interrogate the role that remittances can play in Phuthaditjhaba. To inform our intended research, we seek to draw lessons from evidence on how migration and remittances affect mountainous communities globally. Accordingly, we carry out a systematic literature review (SLR) based on an updated Preferred Reporting Item for Systematic Reviews and Meta-Analysis (PRISMA) 2020 statement supported by bibliometric (co-word) analysis (BA) in VOSViewer. We collected data from the Scopus and Dimensions websites and drew 165 publications, of which only 88 were included after exclusion and inclusion assessments. The PRISMA results show that *Mountain Research and Development*, *Russell King*, and *Nepal* are the most productive and cited journal, the most productive and cited author, and the most researched country, respectively. The bibliometric analysis on keyword co-occurrences revealed that women, agriculture, labour migration, land management, forest, and poverty are the research hotspots. In light of these findings, we proffer important recommendations for future researchers and policymakers and identify thematic research areas for Phuthaditjhaba.

**Keywords:** migration; remittances; mountainous areas; Phuthaditjhaba; systematic literature review; PRISMA; bibliometric analysis; VOSViewer

## 1. Introduction

The United Nations (UN) Agenda 2030, inspired by the pledge "leaving no one behind", has put marginalised societies such as mountainous areas under the spotlight of global sustainable development discourse. Conventionally, people living in poor, lagging regions such as mountains are at high risk of being left behind [1]. Accordingly, achieving inclusive development requires localising sustainable development goals (SDGs) to such areas [2]. Typically, compared with low lands, mountain areas are defined by higher poverty levels [3], more severe food insecurity [4], higher and increasing vulnerability to disasters related to natural hazards, and more severe climate change and variability effects [5]. The lack of economic opportunities [6] exacerbates the situation. Accordingly, mountainous communities opt for migration to address the effects of limited resources and opportunities.

Migration, recognised by the International Organisation of Migration (IOM) as an essential cross-cutting element of the SDGs [7], is hugely expected to significantly drive sustainable development in sending communities. The act of migration has existed since

time immemorial. It is believed to be driven by eight theories, namely, (1) push–pull, (2) neoclassical economics, (3) new economics of labour migration, (4) segmented labour market, (5) network, (6) transnationalism, (7) structural violence and forced migration, and (8) cultural identity theories [8]. All these theories can explain in- and out-migration in Phuthaditjhaba. This emerging mountain border city exhibits a wide range of factors, such as economic, psychological, social, cultural, and political factors, among others. Push factors can be poverty, lack of opportunities, and harsh weather conditions. In contrast, pull factors could be better job prospects, better living conditions, and educational opportunities, especially for people from neighbouring countries and regions, such as Lesotho [9].

Migration can be forced or voluntary. Some structural violence and forced migration happened during the apartheid era when Africans were forced to settle in mountainous homelands such as Phuthaditjhaba [10]. People can take a rational, voluntary decision to migrate to maximise their well-being, taking cognisance of wage differentials, cost of living, and employment opportunities [11]. Labour market structures and inequalities can also drive migration; for example, foreign migrants work in jobs shunned by locals, such as domestic workers, farm workers, or construction workers, among others [12]. Phuthaditjhaba, a border town where people in South Africa and Lesotho share similar cultural values and have relatives across borders, provides a framework for networks and transnationalism where migration helps maintain social, economic, and cultural ties [13]. Remittances sent by migrants to families in their home countries play a pivotal role in improving the welfare of citizens. They are known to be a significant driver of people's mobility [14].

Usually, remittances, mainly financial and social, represent the most significant and direct channel connecting migration to livelihoods. However, mountainous areas such as Phuthaditjhaba, South Africa, have geographical and economic specificities that hinder labour migration gains [15]. For instance, given the rugged terrain, the cost of labour migration is higher. This could include transport, money transfer fees, recruiting agency fees, and government fees and processing charges. Also, because of higher marginalisation, most migrants from marginalised areas are un/semi-skilled and thus obtain low-paying jobs [16]. Thirdly, the rocky and mountainous terrain deters infrastructure development and imposes a cost premium on services [17]. Ultimately, this disrupts the easiness of doing business and provides little return from invested remittances.

Nonetheless, the potential for remittances to impact mountain communities' livelihoods is enormous. Our analysis of data from [18] shows that the top 10 mountainous countries (by area covered by mountains [19]) in the world (see Figure 1) have, on average, the highest share of remittances in gross domestic product (GDP) (20%) compared with all other country groupings. In these countries, mountains cover an average of 83.39% of the total area [19]. Globally, 20% of the land is covered by mountains [19], and remittances contributed only 0.78% of GDP in 2021 [18]. In Figure 1, we make this point very clear. Except for Bhutan, Switzerland, and Macedonia, the more mountainous a country is, the more remittances contribute to GDP. However, we observe that in most of these countries, higher mountain cover and higher remittances in GDP are on average, negatively correlated to GDP/capita. For instance, in Tajikistan, whose territory is 91.9% mountainous (the second largest), the share of remittances in GDP in 2021 stood at 33.4% (the highest). Yet, its GDP per capita is $USD 897 (the lowest). From this brief analysis, we deduce that most mountainous countries are prone to high migration, and, higher remittances, yet they have lower GDP per capita than otherwise. Hence, the current study is motivated to gather evidence on the migration-remittances nexus and sustainability in mountainous areas. This clearly shows the lack of income-generating opportunities in mountainous regions.

The trend in mountainous areas, migration, remittances, and income levels can easily be related to internal migration dynamics for large countries with different geographical landscapes like South Africa. Phuthadijthaba is in the Free State Province, with a contribution to the national GDP of 5.14% in 2017 [20], the second last contributor after Northern Cape (2.19%). Additionally, it's a border town surrounded by better-developed towns

of Harrismith and Bethlehem, just 53 and 70kms away, respectively. The city is home to over 400,000 people with high rates of unemployment- close to 50% in 2019- [21] and characterised by high levels of poverty [22]. All these factors make the city prone to internal out-migration to better-developed provinces and districts. Accordingly, a significant share of its population survives on remittances-local to a more considerable extent, and international to a lesser extent.

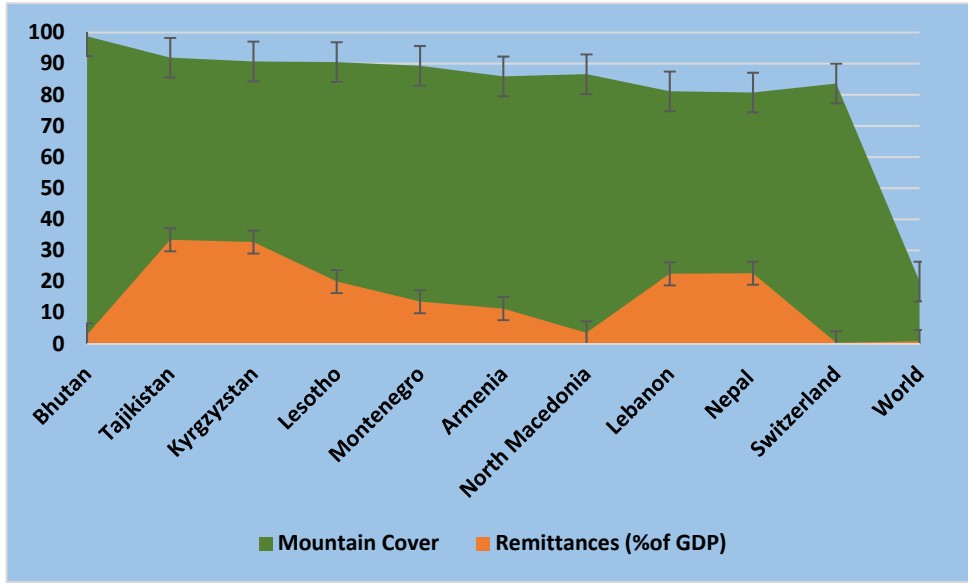

**Figure 1.** Mountain cover as a percentage of total land area and remittances as a percentage of GDP for the top 10 mountainous countries in the world. Source: Authors' compilation from World Population Review (2023) and World Bank Development Indicators (2023).

As shown above, high remittance-receiving countries usually have lower GDP per capita. In 2019, the GDP per capita for Phuthaditjhaba was estimated at USD 1800-00 (Equivalent to 32,400-00 Rands using the average exchange rate of R18:US$1.00.) [23] compared to the national GDP per capita of the whole of South Africa of USD 7055.00 [18]. This clearly shows that Phuthaditjhaba has very low-income generating opportunities. The limited economic opportunities could be a push factor for individuals to migrate to other areas to fend for their families residing in the mountain city of Phuthaditjhaba.

As much as migration can benefit the sending community, concern should be raised against the costs it can bring. As results will show, the sending mountain communities may experience farm labour/skills drain [24], land abandonment [25], reduction in agricultural production and therefore food insecurity [26]. In some ways, migration caused mental health problems for male remitters [27], increased women work burden with no decision-making gains [28], and led to the construction of unplanned and unsecured houses prone to climate change hazards [29].

We take this to raise a question on how remittances affect livelihoods in mountainous communities. This central question has motivated the current study to conduct a systematic literature review on the interplay between migration, remittances, and livelihoods in mountainous areas. We seek to answer the following sub-research questions: (1) What are the major sources of evidence? (2) Who are the prominent authors? (3) Which countries have received more research focus? (4) What are the key thematic areas of research? (5) What are the major trends and findings? and more importantly, (6) What are the research gaps that studies on Phuthaditjhaba (The study area is part of the Afromontane Research Unit (ARU) project titled "Exploring interactions between economic systems, poverty, sustainable development in Phuthaditjhaba, an emerging African Mountain City") could exploit?

In answering these questions, our study seeks to make a significant contribution. While systematic literature reviews have been used to examine the role of remittances on various outcomes [30,31], we couldn't find one focusing primarily on mountainous communities. This is notwithstanding that mountainous areas are distinct and have complex and fragile ecosystems and economies. As such, the reasons for migration, the determination of remittances, how they are used, and their effectiveness in addressing social ills are different. We find some SLRs on mountainous communities in the literature, targeting specific areas such as crop production [32–34]. However, despite its significant role in recipient country GDPs, we could not find one that zero-in on remittances in mountainous areas. As we fill this gap, the evidence from this study becomes critical in providing key stakeholders with a hub of evidence that guides policy on enhancing the sustainability of mountainous areas. Also, it opens more research windows for future studies on the economics of remittances and lives in such areas.

The rest of the study proceeds as follows: Section 2 details the methodology used, Section 3 discusses the major findings, and Section 4 concludes by suggesting areas of further research.

## 2. Methodology

Instead of using the narrative literature review, our methodology benefits from combining two emerging approaches: the systematic literature review (SLR) and bibliometric analysis. While SLR was developed two centuries ago for health science research [35,36], it is only in recent years that its use in social science and business is gaining prominence [37]. A SLR is a literature review methodology used to gather, identify, and critically assess research sources (journals, articles, reports, books, etc.) in an organised manner [38]. Its advantages over traditional literature review are plenty. It uses unambiguous and systematic procedures [39] eliminates author bias [40], and therefore improves transparency [36]. Ultimately, it allows authors to deliver an explicit and comprehensive state of evidence on a chosen topic [41], making it easy to identify and fill research gaps in the literature.

We follow the updated Preferred Reporting Items for Systematic and Meta-Analysis (PRISMA) 2020 Statement to carry out the SLR. As an upgrade to the 2009 Statement, the new version comprises 27-to-do items, with a more robust checklist detailing reporting recommendations, an abstract checklist, and an improved flow diagram (Figure 2) [42]. We use bibliometric analysis (BA) to complement findings from the PRISMA. While the SLR provides an essential guideline on the inclusion and exclusion of research themes, it does not provide a quantitative analysis of the same. That's precisely what BA does. It encloses an array of quantitative techniques (bibliometric analysis such as citation and co-occurrences) on bibliometric units (such as authors, journals, themes) [43].

There are two main BA categories: performance and science mapping. The former involves performance indicators such as publication-related metrics, citations, and number of documents, citation-related metrics, or a combination of both. Science mapping covers citation analysis, co-citation, bibliographic coupling, and co-authorship analysis (see [43] for further details). Researchers should choose the appropriate indicators depending on the specified research objective. Since one of our objectives is to examine the focus of research on remittances in mountainous areas, we used co-word analysis. Distinct from other science mapping techniques, which take cited or citing documents as the unit of analysis, co-word analysis assesses the exact content of the publication [44]. We used the latest VOSViewer 1.6.19 software by [45] to conduct the bibliometric co-word analysis. This allowed us to generate the Network, Overlay, and Density Visualizations, illustrated in Section 3.

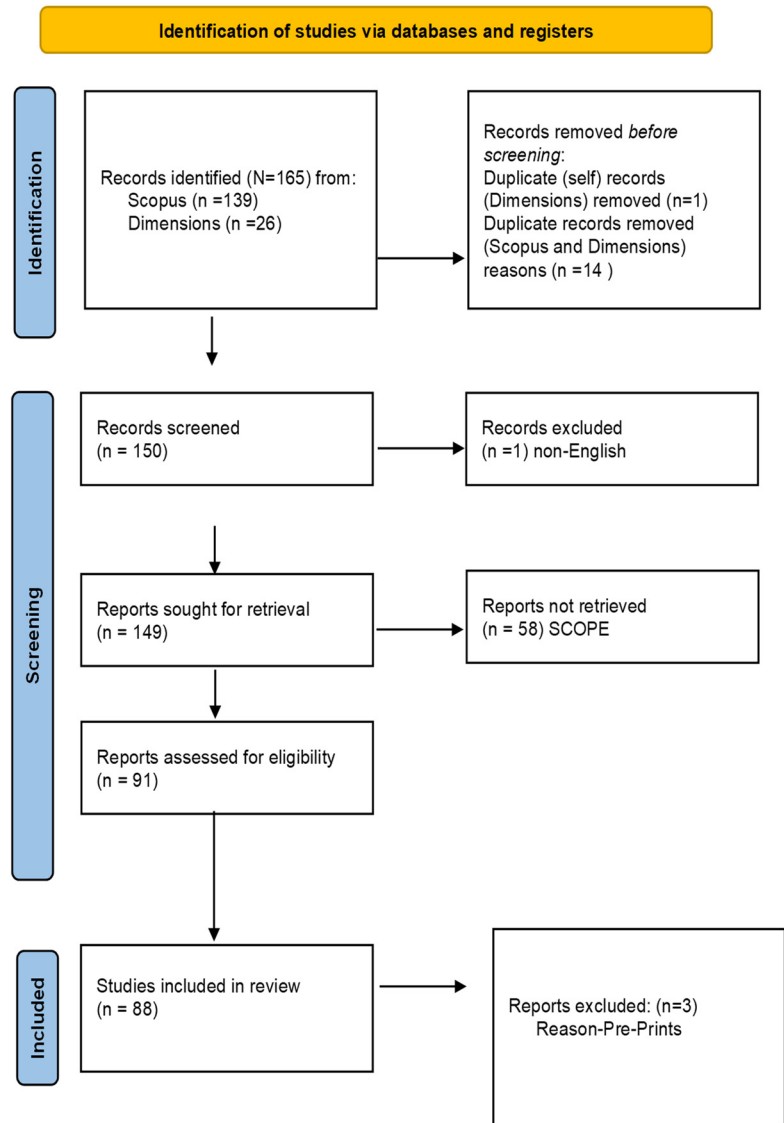

**Figure 2.** Updated PRISMA Flow Chart Statement 2020. Source: Page MJ, McKenzie JE, Bossuyt PM, Boutron I, Hoffmann TC, Mulrow CD, et al. The PRISMA 2020 n71 [42].

*2.1. Data Sources*

We obtained data on publications from two research databases, Scopus and Dimensions. We chose Scopus because it is arguably the most extensive source for bibliometric analysis. At the time of writing, it had 1.8+ billion cited references since 1970, 84+ million records, 17.6+ million author profiles, 94.8+ thousand affiliation profiles, and 7+ thousand publishers [46] Many researchers rely on it for literature reviews due to its quality and trustworthiness [47]. We did not include the Web of Science in our searches due to the close link between the two. A comparative study by [48] documented that 99% of studies indexed in Web of Science are also indexed in Scopus. Hence including both would not significantly improve our search results. Instead, we complement the Scopus search with Dimension, a relatively new database (started in 2018). Dimensions uses different approaches for sourcing data [48], giving us a different dimension. As of 1 April 2023, Dimensions had 134 million documents with 1.7 Billion citations, 12 million datasets, 133 thousand policy documents, and 151 million patents, among other scores [49].

*2.2. Search Strategy*

We used the new advanced search in Scopus using the syntax TITLE-ABS-KEY (migration AND remittances) AND (mountain*). The syntax means that the search targeted the words migration, remittances, and mountain in each paper's title, abstract, and keywords section. These are the three words we investigated in line with the research questions and contribution. The new advanced search criteria in Scopus were used. To broaden our results, we did not constrain the search to any filters at this stage. The search yielded 139 documents comprised of 111 articles, 12 book chapters, ten reviews, two books, and one conference paper. The exact search produced only 26 documents from Dimensions made up of 18 articles, five book chapters, and three pre-prints. We excluded and included according to the PRISMA flow diagram in Figure 2. Considering our research questions and objectives, we retrieved the following from each study: (1) title, abstract, and keywords; (2) authors' documents and citations; (3) sources' documents and citations; (4) country documents and citations.

As can be seen from Figure 2, we followed the three stages: (1) identification, (2) screening, and (3) finalisation. In stage (1), we reported the documents identified from Scopus (139) and Dimensions (26). The source composition is given in Figure 1. In stage (1), 15 duplicates, were removed. In stage (2), we conducted a language check. We removed one document with an English abstract, yet the paper is non-English, and we could not get information from the document. Also, we conducted a detailed assessment of the abstracts, and where we could not decide based on the abstract, we reviewed the paper to check its eligibility. Here, we retained those papers with the three words "migration", "remittances" and "mountain". The process eliminated 58 documents, representing the biggest share (35.15%) of the search results. Of these, 1 (1.72%) did not mention any of the three words, 5 (8.62%) did not include the word remittances, 48 (82.76%) excluded the word "mountain", 4 (6.9%) excluded the words "mountain and remittances". In the last stage, we removed three more documents for the reasons given in Figure 2. We did this for quality purposes, ensuring that only published and complete documents are captured. Finally, 88 papers were considered for analysis.

## 3. Results and Discussion

Our results are in two parts. We start by reporting results on the leading journals, prominent authors, and focal country analysis. In the second phase, we present bibliometric keyword co-occurrence analysis results. At each unit of analysis, we highlight and discuss the key findings and, identify research gaps informing future studies on the interplay between migration, remittances, and livelihoods in Phuthadijthaba. An insight into the themes, methodologies, main findings, and key policy takeaways from the 88 reviewed document are summarised at the end of this section.

*3.1. Main Journals*

From the Scopus database, the leading journals focusing on the interplay involving migration, remittances, and mountain issues and documents' h-index are given in Figures 3 and 4a. They provided 24 (29.6%) of the final documents in Scopus. The leading journal is Mountain Research and Development, with the most documents, 6 (7.4%) of the included documents. It also recorded the highest citation (128). This is followed by the Journal of Ethics and Migration 4 (4.9%). These results can benefit future research interests on the interplay between migration, remittances, and sustainability issues in Phuthadijthaba, an emerging and marginalised mountainous City. Firstly, more information can be obtained from these top journals. The MRD is the most relevant source because, unlike other journals, it is centred on mountain life. Its three peer-reviewed sections, Mountain Development, Mountain Research, and MountainAgenda, zero in on "Transformation Knowledge", "Systems Knowledge", and "Target Knowledge" aspects [50]. As further results will show, it speaks more on migration, remittances, and outcomes of interest in mountainous areas. Figure 4a shows that the documents have a h-index of 27. As such,

at least 27 journals have been cited 27 times. This indicated high productivity and broad audience of the journals.

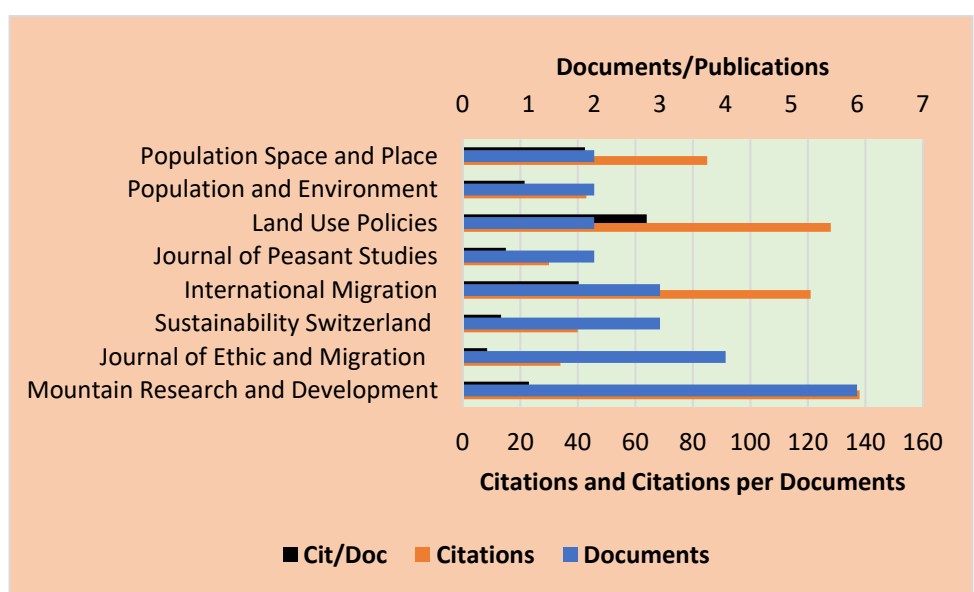

**Figure 3.** Journals documents (secondary axis), citations, and citations per document. Source: Authors' compilation from Scopus search output.

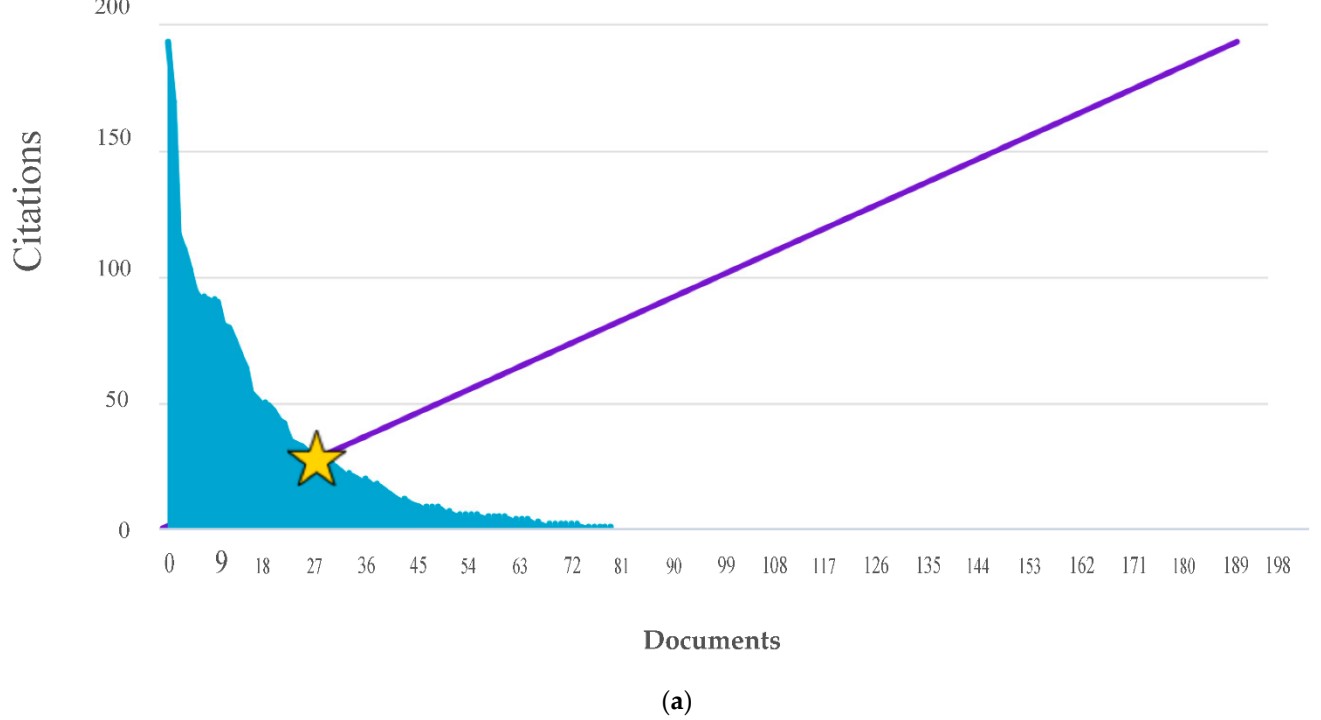

**Documents**

**(a)**

**Figure 4.** *Cont.*

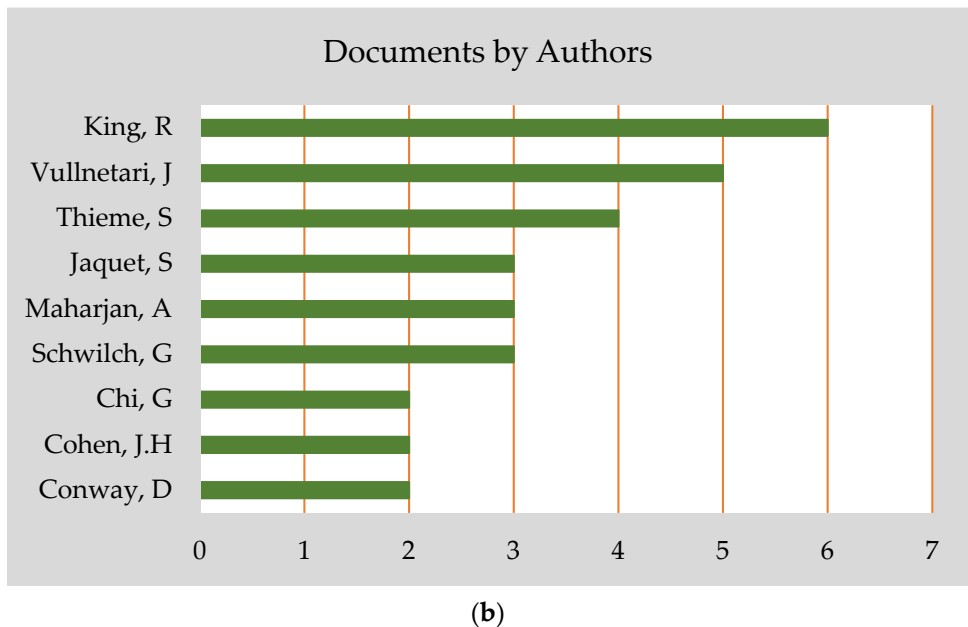

**(b)**

**Figure 4.** (**a**) Journal's documents h-index. Values to the left of the star and above the 45-degree line (in purple) show combinations in which documents are less than citations. Points below the star and to the right of the line show more documents than citations. The star indicates equilibrium where the number of documents (27) are cited as much. (**b**) Documents by author. Source: Scopus search results analysis.

### 3.2. Main Authors

The results of the author analysis are illustrated in Figure 4b. The prominent author is King, who collaborated in 6 publications [51–56]. All the documents take Albania as a migrant sending country and focuses on the gender and family aspects of remittances among the recipients, mainly women and the elderly. Out migration of people in mountain communities like Phuthaditjhaba can result in elderly, children and women remaining behind while other groups migrate to find better opportunities elsewhere. From the literature we deduce that the role of remittances is guided by gender and patriarchal issues. Also, in [51,54,55] the role of remittances in cushioning sending households from social isolation and loss of intimate-based trans-generational care and family relations is appreciated.

Ref. [51] confirmed the poverty alleviation role of remittances. Only one, Ref. [55] emphasised on internal migration, as others focus on international migration. While King's work is limited in terms of lack of heterogeneity (by concentrating on Albania), and in time-space, we draw important takeaways. Firstly, Albania becomes a potential candidate for comparable research from which Phuthaditjhaba can benefit. Secondly, we pick that the role of internal migration is side-lined even by top scholars in the field, irrespective of its dominance and potential. From this perspective, considering the economic impact of internal migration and remittances becomes an area of interest.

### 3.3. Country Analysis

Results on country coverage are illustrated in Figure 5a. The leading country associated with research on migration, remittances and mountain communities is the United States of America (USA), followed by Nepal, receiving 25 and 14 document mentions, respectively. We find USAdominance not surprising. As has been the case in the past 40 decades, it remains the principal destination of migration. As shown in the IOM World Report on Migration, in 2020, the USA was home to 51 million migrants, with Germany a distant second with 16 million [57]. Given its proximity to Central American countries, most

of which are mountainous, it's logical that many studies have examined how receiving countries are impacted.

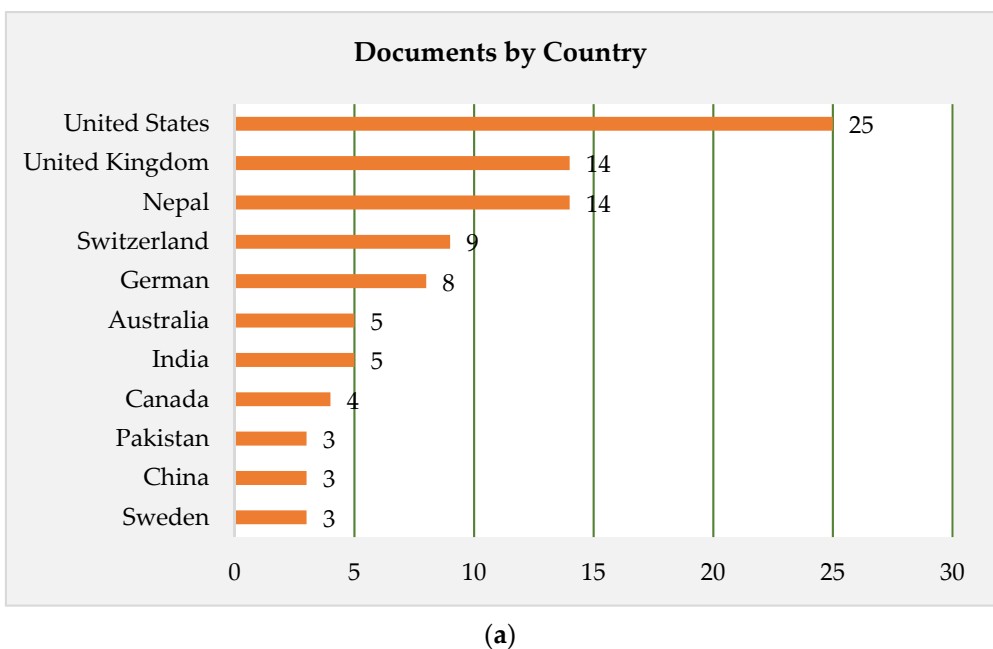

(**a**)

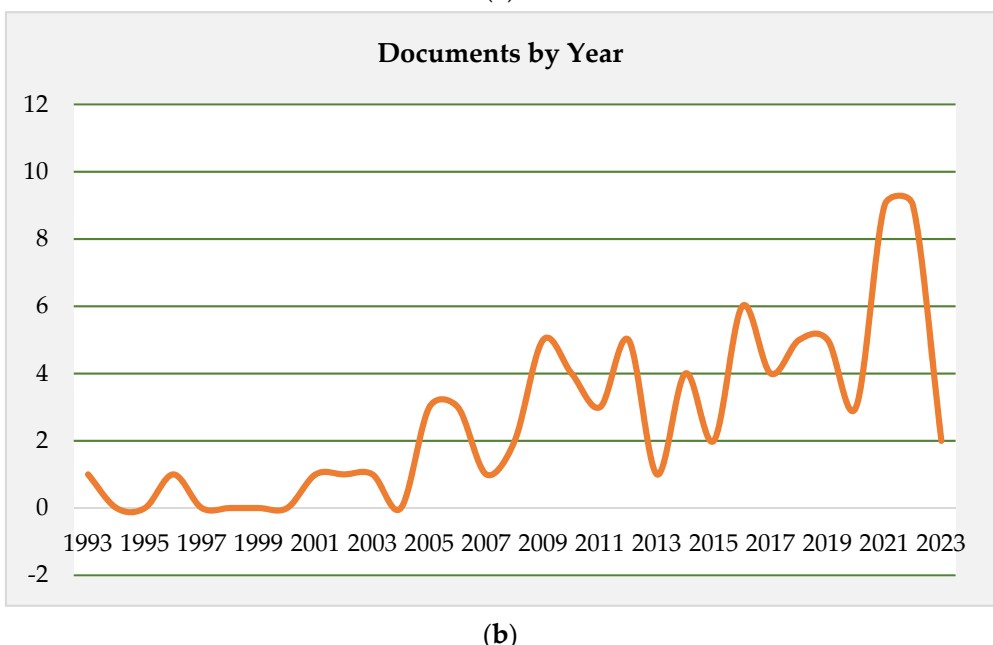

(**b**)

**Figure 5.** (**a**) Documents by country. (**b**) Documents by year. Source: Scopus search results analysis.

Since the impact of migration is mainly on sending countries, we find Nepal to be a more exciting case than the USA. The 14 documents on Nepal mainly focus on the effects of migration and remittances on agriculture systems [25,58,59], forestry systems [60,61] or both [62], and land management systems [63,64]. Evidence by [25] shows that remittances positively impact maize yield and food security. [58] associate the positive impact of migration and remittances with its technology transfer effect.

We also picked that remittances play another vital role in mountainous communities: diversification of recipient household income. Refs. [60,65] show that remittances reduces reliance on forest and agriculture income. These are the two major sources of income in mountainous areas and the worst affected by climate change. Phuthaditjhaba has

pockets of agriculture activities, real estate and infrastructure development and other lively activities [66,67]. Investigating the impact of migration and remittances on such activities provides valuable insights.

Accordingly, the role of remittances in climate adaptation strategies cannot be overlooked in mountainous communities such as Phuthaditjhaba. While other mountainous areas are endowed with forestry resources they can count on, Phuthadijthaba mountains are virtually treeless grasslands. Accordingly, there is no opportunity for remittance recipients to invest in the timber industry. At the same time, the findings here suggest a potential area of investment that can go a long way in reducing poverty and climate change vulnerability. Phuthaditjhaba experiences harsh climatic conditions ranging from frigid temperatures, snow, hailstorm, flash floods and windy conditions [67]. It is important to understand the role remittances play towards resilience against these harsh climatic conditions of Phuthaditjhaba.

In addition, migration and remittances are found to have environmental sustainability effects. Ref. [63] show that remittances are mainly used for food and goods and less for agriculture. As a result, pressure on land is reduced. Also, internal movement to low-lying lands and marketplaces often leads to land abandonment, which may comprehend environmental sustainability. This is complemented by [61], who found that remittances have led to increased forest regeneration in Nepal.

The takeaway point from these studies is that emerging transmission mechanisms like technology transfer must be explored in analysing remittances' effects. Also, we acknowledge the growing role remittances play in climate adaptation strategies in mountainous areas. These findings are important in analysing the interplay between migration, remittances, and sustainability in Phuthaditjhaba. However, less attention has been given to poverty eradication issues and the transition to green energies. While in the former, one argument can be the differences in the poverty levels (high in Phuthaditjhaba), the same cannot be said about the latter. As global environmental sustainability becomes the flagship policy focus, examining the role of remittances in renewable energy transition can help shape the same in mountainous areas.

Another finding we note from the country analysis is the dearth of such African studies. Only three Ethiopia, Kenya, and Morocco countries are mentioned in two studies [68,69]. Ethiopia and Kenya feature in a comparative study with Nepal [68]. The study examined the interplay between cyclical labour migration and agrarian transition in the three countries. The major finding is that migration is mainly motivated by capital market expansion with mediation from domestic cultural, political, and ecological dynamics.

Furthermore, as in most of the evidence already reviewed in this paper, remittance income is associated with increased agricultural productivity growth on the one hand and increased land burden on the other. The lack of studies from Africa requires a shift of focus. Most studies from the findings have shown that remittances are connected to various forms of livelihoods and climate-related dynamics in mountainous areas. The case for Africa, and indeed Phuthadijthaba, carries more weight considering higher levels of poverty and climate change vulnerabilities.

### 3.4. Documents over Time

The findings on documents by year (Figure 5b) indicate the growing focus on mountainous areas over time. From 1993 to 2004, only five documents were picked from our search. The first document [70], coincided with a huge step by the United Nations towards recognising mountain communities' sustainable development in 1992. As part of the Agenda 21 of the Conference Environment and Development, the document "Managing Fragile Ecosystems: Sustainable Mountain Development" brought to shore mountain areas as fragile ecosystems requiring special recognition in development circles [71]. Since then, the attention to the importance of mountains has a number of global initiatives that continue to put mountain development on the radar. The United Nations General Assembly declared 2002 the International Year of Mountains [72], and later, on 11 December of every

year, as the International Mountain Day [73]. The day has been set aside to create awareness about the importance of mountains to life and to highlight the opportunities and constraints in mountain development, among other initiatives. Recently, 2022 was designated as the International Year of Sustainable Mountain Development. The increase in research on migration and remittances is testimony to this. As seen from Figure 5b, there has been an increase in publications since 2004.

### 3.5. Key Words

To generate a deeper analysis of the main targeted aspects, we further carried out a bibliometric analysis of the keywords from documents in Scopus. Results are shown in Figures 6–8. Figure 6 shows the Network Visualization showing the clusters, co-occurrences of words, number of links and strength of links. There are seven clusters represented by different colours. For example, red represents cluster 1, with 25 items. The network was generated from VOSViewer software 1.6.19 [45]. The Network Visualisation shows the heatmap of the keywords mentioned at least five times in all the studies under our literature review.

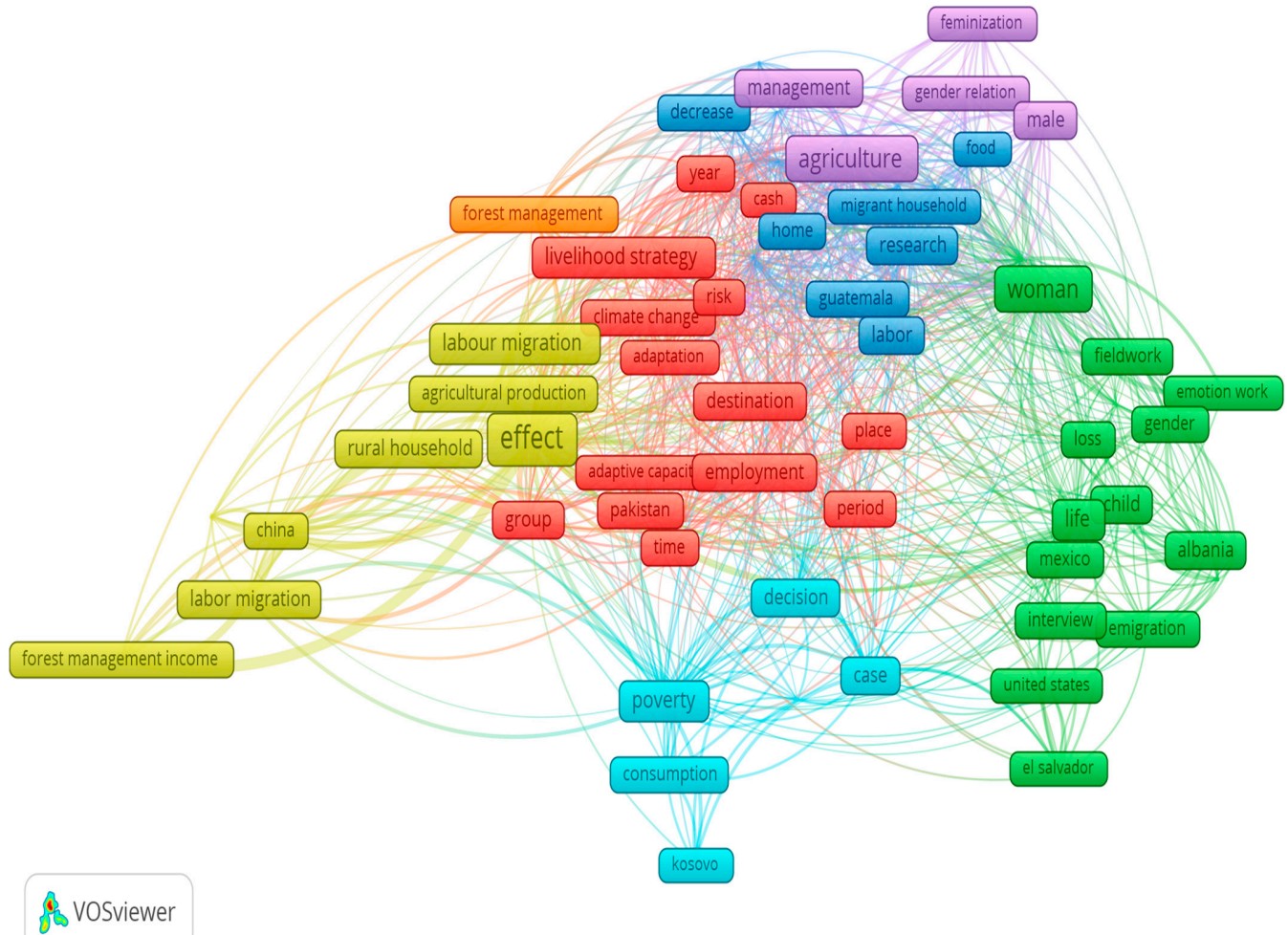

**Figure 6.** Keyword co-occurrences Network Visualization. Source: Generated from VOSViewer.

The bibliometric keyword co-occurrence results are explained by five visualisation elements: the item, link, strength, network, and cluster. An item is the unit of analysis, in our case keywords. A link is a path connecting two terms (key words) with the connection measured by some positive value showing its strength [45]. The link and strength in Figure 6, capture the number of documents/publications in which two keywords co-exist. A collection of items and links gives the network. Lastly, a cluster or community

shows a group of strongly related items. The keywords are shown in nodes, whose size and label show their weight. Also, the distance between nodes indicates the keywords' relatedness [45].

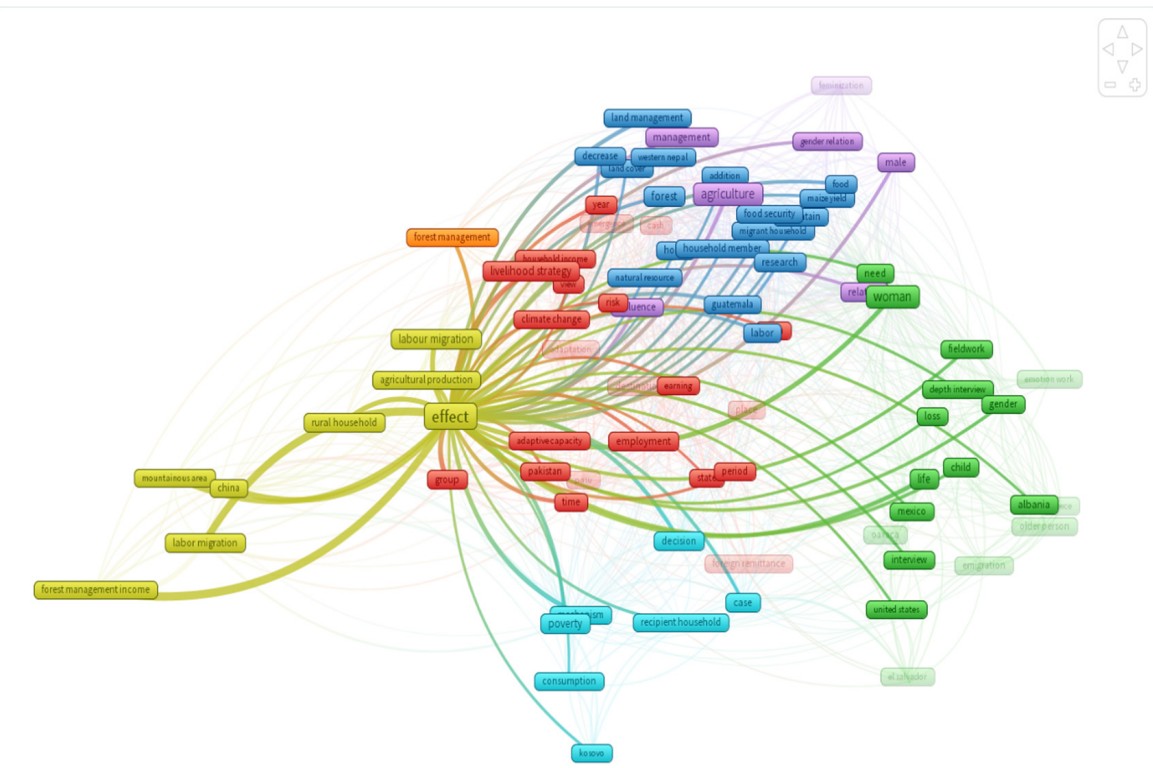

**Figure 7.** Effect visualisation. Source: Generated from VOSViewer.

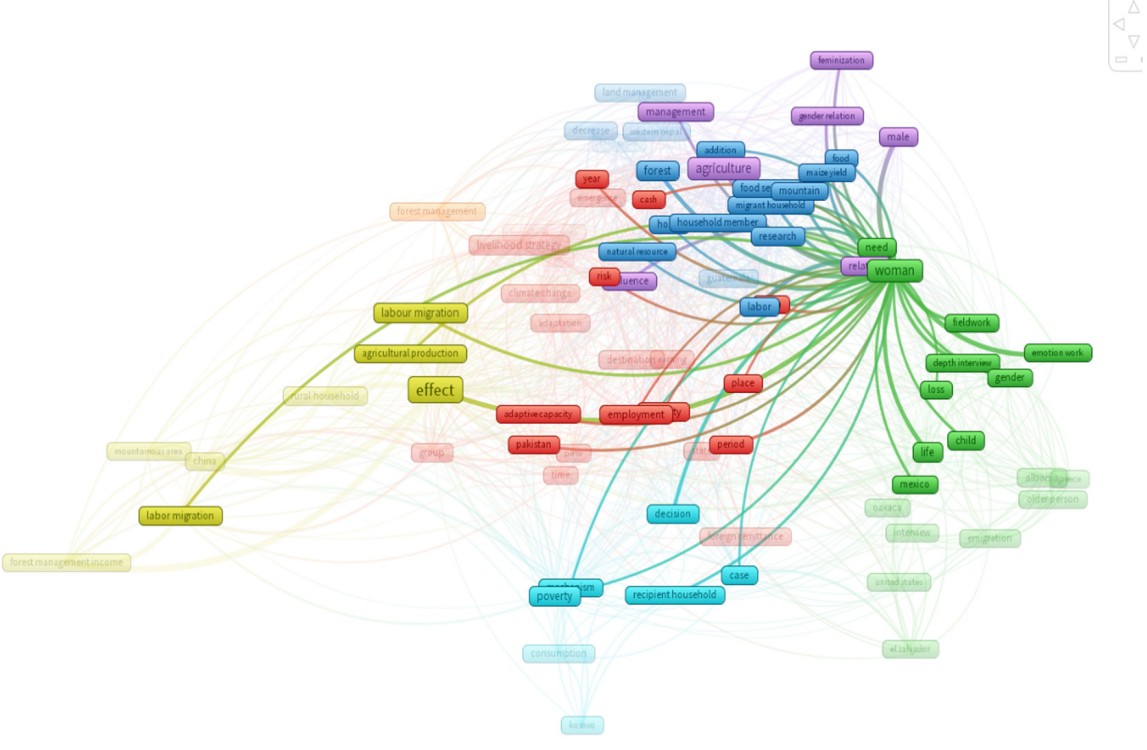

**Figure 8.** Women visualisation. Source: Generated from VOSViewer.

Accordingly, it can be observed from Figure 6 that the word effect has the biggest node, having been mentioned at least five times in 77 publications. (These values are obtained from zooming the data in VOSViewer 1.6.19). The files are attached in Supplementary Materials. One must install VOSViewer (freely downloadable at www.vosviewer.com, accessed on 28 May 2023). The word has 69 links with a strength of 1036 (see Table 1). In the vicinity of the effect node are the words agricultural production, rural household, labour migration, and adaptive capacity. The results suggest that most studies examined the effect of migration and remittances on various issues affecting mountain communities. Effect analysis was stronger on the nearby nodes. The second biggest node is on "women", with 41 co-occurrences and 50 links with a strength of 437. Its neighbourhood comprises the words gender, feminisation, child, male, food security, male relations, and migrant household, among others. As shown in Table 2, gender issues are the most researched, with 23 (26%) of the documents under review. These issues are central to sustainability as they are connected to the SDGs.

**Table 1.** Keyword (top 10) co-occurrence bibliometric analysis results.

| Key Word | Co-Occurrences | Links | Links Strength | Cluster |
|----------|----------------|-------|----------------|---------|
| Effect | 77 | 69 | 1036 | 4 |
| Women | 41 | 50 | 437 | 2 |
| Agriculture | 39 | 63 | 585 | 5 |
| Labour migration | 23 | 43 | 264 | 4 |
| Management | 12 | 46 | 232 | 5 |
| Forest | 22 | 44 | 266 | 3 |
| Poverty | 20 | 42 | 256 | 6 |
| Life | 18 | 54 | 232 | 2 |
| Environment | 17 | 38 | 177 | 3 |
| Albania | 16 | 23 | 157 | 2 |

Source: Authors' compilation from bibliometric analysis results.

The findings here show that women are a focal group in the migration-remittances-mountain relationship. This area was mainly covered by [51,52,56] among others. The main findings from these studies were discussed under author analysis. Food security, children, and loss are close by to reflect the vulnerability of women to migration. It has been shown that women are less likely to migrate and are left to take care of the children, are exposed more to loss of social intimacy and face food insecurity [25,63,65,74]. It is critical to explore how migration and remittances impact women and their economic role in Phuthaditjhaba. Zooming the "women" node (in VOSViewer) also shows that "labour migration" is the furthest connection, signalling relatively low labour migration than males [65]. The rest of the research hotspots are shown in the co-occurrence statistics in Table 1. To elaborate on the visualisation of the words in Table 1, we show the illustrations for "effect" and "women" in Figures 7 and 8.

The results we presented in Table 1 are of great importance to scholars, funders, and research institutions. If remittances are to propel the livelihoods of people in mountainous communities, then more effort should be put into the hotspots to maximise the return and minimise the costs of migration and remittances. Nonetheless, looking at Table 1, we make an important observation. Emerging issues that stakeholders should focus on are not received deserving attention. In particular, the transition to renewable energies has become an increasingly important feed into the global efforts to combat climate change. Mountainous communities are associated with energy poverty [75]. In its Renewables Readiness Assessment of the Kingdom of Bhutan, a mountainous country, the Ref. [76] revealed that despite the terrain, the potential for generating renewable energy from solar and wind is vast. Key to the exploitation of the huge potential is financing.

**Table 2.** Summary of research focus, methodology, key findings, and policy takeaways.

| Migration, Remittances, and Climate Change Vulnerability, Adaptation, and Resilience | | | | |
|---|---|---|---|---|
| Author(s) | Research Focus | Methodology | Key Findings | Key Policy Takeaways |
| **1**. [77], **2**. [78], **3**. [79], **4**. [65], **5**. [80], **6**. [81], **7**. [82], **8**. [83] | > Migration, climate and change resilience<br>> Remittances, natural resource extraction, and management<br>> vulnerability, Floods, Water hazards<br>> Household adaptation, and climate change | > Household survey<br>> Literature review<br>> Case study<br>> Difference in Difference Approach | > Capacity building on financial literacy significantly impacts climate resilience strategies<br>> Remittances lead to more extraction of natural resources for less capital-constrained households<br>> Reducing dependence on agriculture<br>> Decrease in agriculture production.<br>> Recipients are less vulnerable, have lower dependence on the environment, are more financially included, and are more resilient to food insecurity | > Climate resilience policy should target recipients<br>> Increase training of state and non-state actors on proper utilisation of remittances.<br>> Need to promote rural labour markets and educate asset-poor households<br>> Provide incentives for remittance invested into climate change resilience strategies<br>> Investing remittances in diversified income sources is key to sustainability. |
| Migration, Remittances, Agriculture, and Land and Forest Management | | | | |
| Author(s) | Research Focus | Methodology | Key Findings | Key Policy Takeaways |
| **9**. [84], **10**. [60], **11**. [85], **12**. [68], **13**. [61], **14**. [86], **15**. [87], **16**. [59], **17**. [88], **18**. [89], **19**. [90], **20**. [91], **21**. [63], **22**. [64], **23**. [92], **24**. [93], **25**. [94], **26**. [95], **27**. [27]. | > Forest management<br>> Agrarian transition<br>> Agroforestry<br>> Land cover and crop production<br>> Firewood consumption | > Household surveys<br>> Interviews<br>> Focus group discussion<br>> Panel-Tobit ([84]<br>> Qualitative mapping method<br>> Structural Equation Model (SEM) [87]<br>> Multinomial logistic regression model [91] | > Negative impact on forest management income, remittances have a promoting effect.<br>> Agriculture intensification, land abandonment, and reforestation.<br>> Remittance supports agriculture which remains the primary livelihood strategy [90]<br>> Remittances key in the adaptation of agroforestry practice<br>> Remittances were mainly used for food and less for agriculture [63].<br>> Remittances flow support land ownership increase livelihoods without environmental degradation [93]<br>>Against expectations, migration is leading to forest loss [27]. | > Need to improve infrastructure and training to support farmers.<br>> Government, local authorities, and communities need to integrate remittances into forest management and agricultural transformation.<br>> Remittances should not be the basis for land abandonment but should be used to increase agricultural productivity |

**Table 2.** *Cont.*

| Migration, Remittances, and Gender Issues | | | | |
|---|---|---|---|---|
| **Author(s)** | **Research Focus** | **Methodology** | **Key Findings** | **Key Policy Takeaways** |
| **28**. [96], **29**. [58], **30**. [97], **31**. [25], **32**. [98], **33**. [99], **34**. [62], **35**. [100], **36**. [101], **37**. [102], **38**. [103], **39**. [104], **40**. [105], **41**. [106], **42**. [107], **43**. [108], **44**. [28], **45**. [109], **46**. [51], **47**. [110], **48**. [111], **49**. [52], **50**. [112], **51**. [25]. | > Rural out-migration, Feminization of Agriculture<br>> Non-migrant women, and emotional work<br>> Gender roles, farming system<br>> Bride kidnapping<br>> Male out-migration, Masculinities<br>> Family separation, stress<br>> Transnational care | > Household surveys<br>> Interviews<br>> Focus group discussion<br>> Literature review | > Remittances shape the struggles over agricultural, water, and land resources.<br>> There are emotional costs and inequality in communication among non-migrant recipients<br>> Kidnap households are more likely to receive remittances, which translates to unified, patriarchal households.<br>> Women shouldering additional work burden, leading to reduced farm production and increased food insecurity<br>> Marginally improvement in women's empowerment and leadership [106]<br>> Migration causing depressive clinical problems. Separating single and married men were vulnerable to poor mental health [108]<br>> Despite women's increased role in farming, there is no change women's decision making [28]<br>> Migration increase responsibilities of women, yet they can't make independent choices against prevailing gender norms [111] | > Remittance income is necessary but insufficient to achieve women's empowerment<br>> Using remittances in educating the girl child provides more empowerment gains<br>> Allowing mountain women to be active labor migrants can increase migration gains in their societies.<br>> Recipient women need to invest remittances to reduce dependence and diversify income to increase asset ownership<br>> There is need to integrate gender policy with migration, agriculture, and food security policies.<br>> Accounting for women choices and preferences in developing and designing agricultural interventions is key for the sustainability of mountain farming systems. |

**Table 2.** *Cont.*

| Author(s) | Research Focus | Methodology | Key Findings | Key Policy Takeaways |
|---|---|---|---|---|
| **Migration, Remittances, and the Built Environment** | | | | |
| **52** [29], **53**. [113], **54**. [114]. | > Remittances and house boom<br>> Housing, and disaster<br>> Built environment of migration | > Literature review, multi-methods approach<br>> Household surveys<br>> Census data | > Increased remittance incomes caused a housing construction boom, albeit without urban planning or infrastructure<br>> Higher risk and vulnerability from unstandardized housing<br>> Remittances instrumental in building houses, temples, schools, shops, roads, and bridges. Many are now heritage items [114] | > Government and non-governmental organization training on incorporating remittances in urban planning programs.<br>> Planning, capacity building, risk awareness, diaspora engagement |
| **Migration, Remittances, and Wellbeing (Consumption, Poverty, Expenditure, Food Security)** | | | | |
| **55**. [115], **56**. [116], **57**. [117], **58**. [118], **59**. [119], **60**. [120], **61**. [74], **62**. [26], **63**. [121], **64**. [122], **65**. [123], **66**. [124], **67**. [125], **68**. [70]. | > International remittances and health outcomes<br>> Remittances and expenditure patterns<br>> Livelihoods, Wellbeing, poverty<br>> Food security<br>> Socioeconomic status | > Living Standards Survey (TLSS)<br>> In-depth interviews and focus group discussions<br>> Instrumental variable estimation [115]<br>> Panel data methods [116]<br>> Generalized Least Squares estimator [121]. | > Remittances significantly improve health expenditure and welfare [115].<br>> Little effect on household spending, changes in them cause marginal changes in the budget shares of food and medical expenses [116].<br>> Negative effects on food security due to decrease in agriculture production [26].<br>> Neither local nor international remittances positively affect investment expenditures [126]. | > Remittances are volatile and should be complemented with public health system investment [115].<br>> Investing remittances in non-agricultural and non-migratory livelihoods can enhance food security in the long run.<br>> There is need for a proper strategy to combine remittances and animal husbandry for livelihoods and environmental sustainability.<br>> If remittances are spent with no investment, they cat only as a short term coping strategy. |

**Table 2.** *Cont.*

| Migration, Remittances, and Cross-Cutting Issues | | | | |
|---|---|---|---|---|
| **Author(s)** | **Research Focus** | **Methodology** | **Key Findings** | **Key Policy Takeaways** |
| **69.** [127], **70.** [128], **71**. [53], **72.** [69], **73.** [129], **74**. [130], **75.** [131], **76.** [132], **77.** [133], **78.** [134], **79.** [135], **80.** [136], **81** [137], **82.** [24], **83.** [138], **84.** [139], **85.** [140], **86.** [141], **87.** [142], **88.** [143]. | > Out-migration, COVID-19, and returning migrants. > Modernization, migration and translocality > Educational remittances > Industrialization > Community organization > Labour migration trends > Remittance networks and geopolitical shocks > Conflict | > Household surveys > Ethnographic research [141] > Generalized least squares (GLS) [136] | > COVID-19 forced returnees more likely to migrate back to sustain livelihoods through remittances. > Migration strategies, through sectoral and spatial livelihood diversification, helped in translating external modernization into mountain development. > Migration can have negative effects (the misery it produces, growing urban poverty or positive, on international migration and remittances [69] > Strong forms of community organization can make the difference between migration contributing to underdevelopment and development [129] > Remittances to the NIS are more vulnerable to geopolitical and economic shocks affecting the Russian Federation [136] | > Providing social amenities e.g., educational and health infrastructure and services, can cab out-migration. > Low earning migrants can still contribute to the development of their origins through ideas, new methods, technology. > Communities can use the organizational capacity of traditional governance systems to access remittances from migrants for the benefit of the community as a whole. |

Source: Authors' compilation from PRISMA results.

It will be interesting, therefore to analyse the role remittances can play in renewable energy transition in mountainous areas. In Phuthadijthaba, such an investigation is even more important for one major reason. Energy poverty is exacerbated by the unavailability of alternative energy sources due to the absence of woodlands. In addition, the city experiences long hours of electricity load shedding. Also, there is minimal uptake of solar energy, while no wind energy infrastructure exists. Accordingly, finding out how remittances can contribute to clean energy sustainability in Phuthadijthaba becomes imperative.

We also generated variants of the Network visualisation in Figures 9 and 10. The former is a Density Visualisation, and the latter is an Overlay visualisation. This provides another angle into our analysis. In Figure 9, every point in the visualisation has a colour that shows the density of keywords at that point. The colours range from blue (lowest weight) to red (highest weight). It can be seen that the keywords with the highest co-occurrence, links, and link strength in Table 1 are in the red zones. This identifies them as hotspots of research.

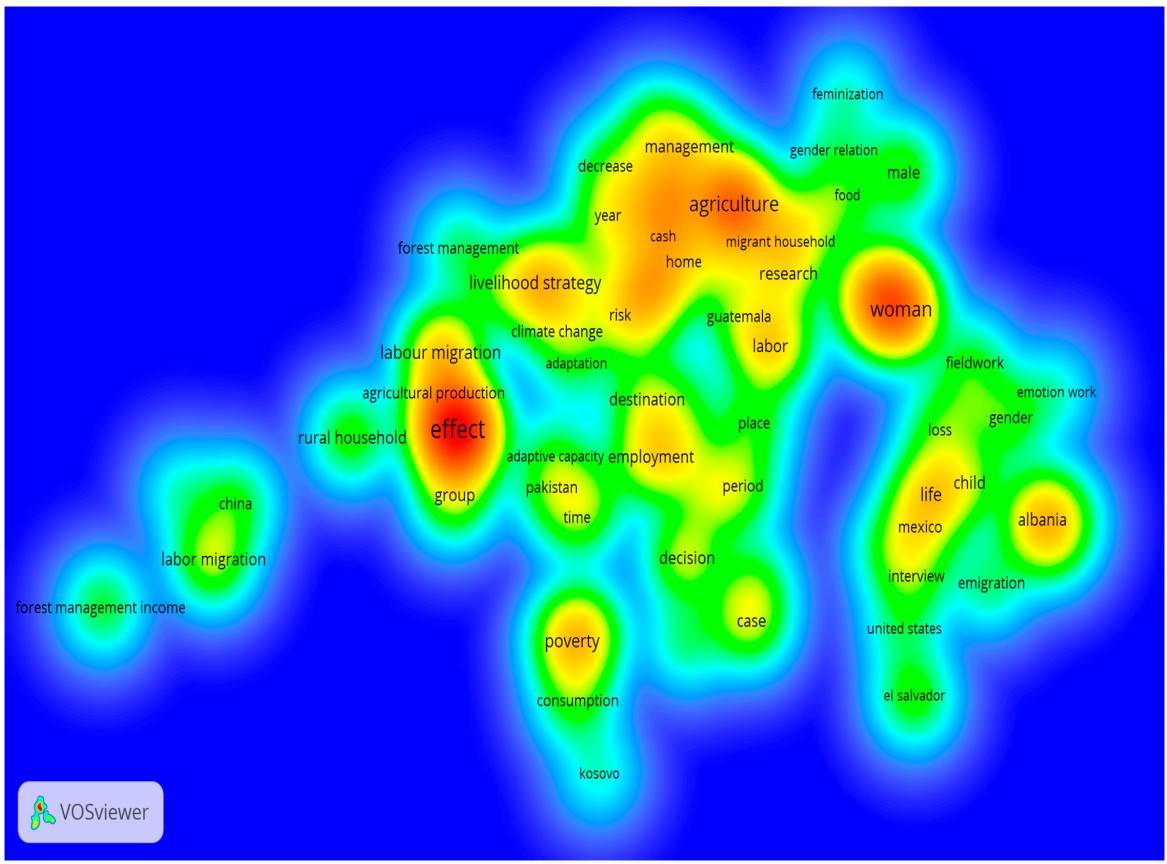

**Figure 9.** Density visualisation. Source: Generated from VOSViewer.

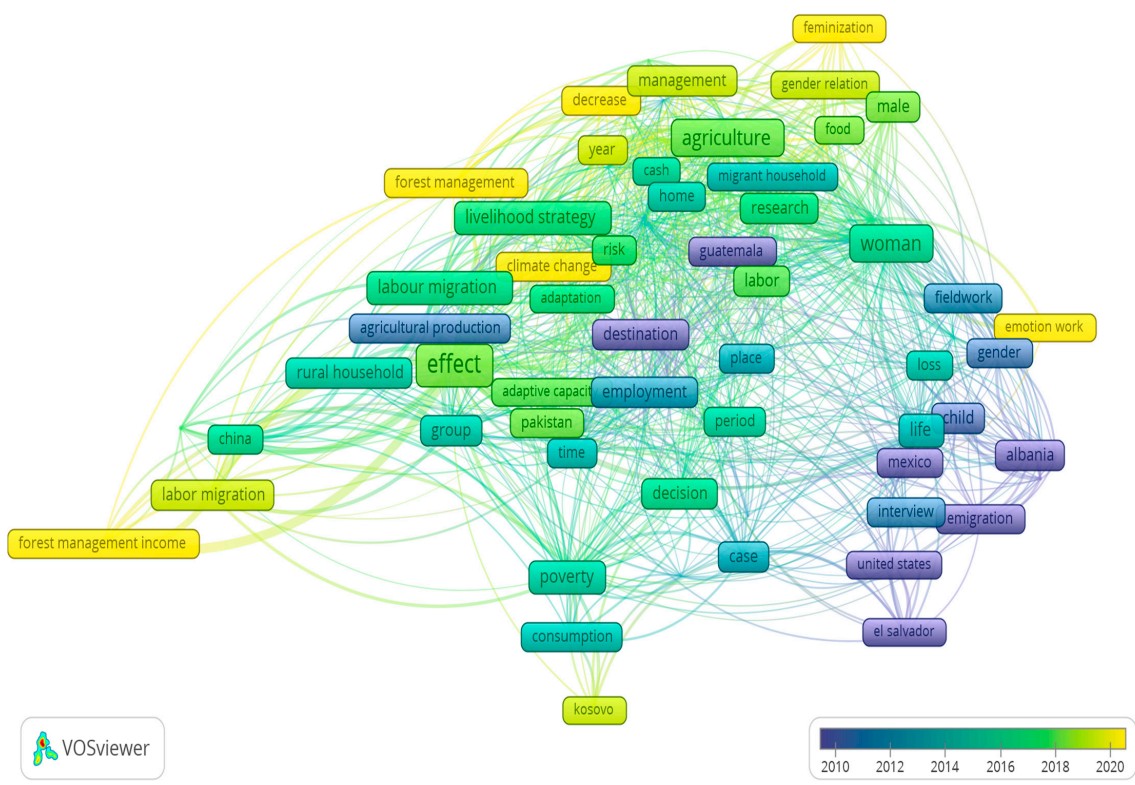

**Figure 10.** Overlay visualisation. Source: Generated from VOSViewer.

Similarly, in Figure 10, the Overlay visualisation reveals the scores for each keyword, with the size proportional to the colour (blue lowest, yellow highest). We related the keyword score to years, as shown in the key bar. From this, we can see the emerging trends in studies over time. For instance, the focus has been moving from determining migration and foreign remittances to effects on land management, gendering issues, and food security, among others.

Lastly, we summarise the insight on the themes, methodologies, main findings, and key policy takeaways from the 88 reviewed documents in Table 2. We divided the research focus into six themes: (1) climate change vulnerability, adaptation, and resilience; (2) agriculture, land and forest management, (3) gender issues, (4) the built environment, (5) well-being (consumption, poverty, expenditure, food security), and (6) cross-cutting issues. Gender issues are the most researched, with 23 (26%) of the documents, while the built environment 3 (3%) is the least researched area. The summary of findings in Table 2 shows that migration generally benefits sending communities. However, we picked some costs. These occurs when they experience farm labour/skills drain [24], land abandonment [25], reduction in agricultural production and therefore food insecurity [26]. In some ways, migration caused mental health problems to male remitters [27], increased women work burden with no decision making gains [28], led to construction of unplanned and unsecure houses prone to climate change hazards [29].

## 4. Conclusions and Areas of Further Research

In light of these findings, we proffer important recommendations to researchers and policy makers with interests in migration, remittances and mountainous communities. We then suggest areas of research targeting Phuthadijthaba. Our findings on source documents reveals that Mountain Research and Development (MDR) journal is the most productive. As such researchers can rely on it for literature review and possible publication of their work. From author analysis, we find King Russel to be the main author. However, all his research were on Albania and the recent captured publication is over a decade old. Even so, drew important takeaways: (1) Albania becomes a potential candidate for comparable research from which Phuthadijthaba can benefit from (2) we pick that the role of internal migration is side-lined even by top scholars in the field, irrespective of its dominance and potential. From this perspective, considering the economic impact of internal migration and remittances becomes an area of interest.

Country analysis showed that Nepal is the most researched, and our studies on Phuthadijthaba can benefit from trans-local research. However, we note that less attention has been given to poverty eradication issues and transition to green energies. While in the former, one argument can be the differences in the poverty levels (high in Phuthaditjhaba), the same cannot be said about the former. As global environmental sustainability becomes the flagship policy focus, examining the role of remittances in renewable energy transition can help shape the same inmountainous areas.

Lastly, bibliometric co-word analysis highlighted gender, feminisation, child, male, food security, male relations, and migrant households as the most studied areas. Researchers seeking to evaluate the impact of remittances can concentrate on these issues. Equally, organisations and governments can also focus on funding, expenditure, and policy aspects on these hotspots. In a nutshell, forthcoming studies on the role of remittances in Phuthadijthaba will follow the themes in Table 2. It is also vital to examine remittances' role in the renewable energy transition. These findings should be considered within the context of the following shortcomings. The documents reviewed in this study are from the major databases only. While maintaining quality is critical, broadening the scope to include more literature sources such as Google Scholar can broaden the evidence.

**Supplementary Materials:** The following supporting information can be downloaded at: https://www.mdpi.com/article/10.3390/su151914621/s1. VOSViewer Network Visualization.

**Author Contributions:** Conceptualization, R.S. and C.M.; Methodology, R.S.; Software, R.S.; Validation, C.M.; Resources, C.M.; Writing—original draft, R.S.; Writing—review & editing, C.M. All authors have read and agreed to the published version of the manuscript.

**Funding:** The authors were supported through a South African Department of Science & Innovation and National Research Foundation (DSI-NRF) Risk & Vulnerability Science Centre award to the Afromontane Research Unit (ARU, University of the Free State; Grant No. 128386).

**Institutional Review Board Statement:** Not applicable.

**Informed Consent Statement:** Not applicable.

**Data Availability Statement:** Data used in the study is made available on reasonable grounds.

**Conflicts of Interest:** The authors declare no conflict of interest.

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
