# Peer review of "A Systematic Literature Review on Migration and Remittances in Mountainous Regions: Key Takeaways for Phuthaditjhaba, Free State, South Africa"

_sustainability, doi:10.3390/su151914621_

Round 1

Reviewer 1 Report

Interesting and attractive topic, even if not all adequate to this journal, but it can be better shown the compatibility with the ”Sustainability” Journal goals.

Is need to emphasize clearer what is pursued through the work, respectively the purpose and objectives of the paper - clearly highlighted.

The motivation and argument for choosing the topic which is it?

What are the criteria by which you chose the keywords for the analysis?

The degree of novelty of the work- is not highlighted.

The argument for the delimitation of the region chosen for the analysis.

All figures require personal opinion with interpretation and explanation of their role in the research.

What is the usefulness of figures 6, 7, and 8 in the work?

Poor literature and references for a review literature analysis.

It is not shown exactly the basic and consecrated literature for the topic analyzed.

The results and discussions should be better highlighted, accentuating the author's contribution.

The utility and limits of the paper must be presented in ”Conclusions.”

Author Response

Thank you so much for the review, the comments, and the insights. We have attached a table showing our responses to all the comments and suggestions.

Reviewer 2 Report

This article provides a nice synthesis of some literature linking the issues of migration and remittances. The article is scholarly and very innovative on the topic. The conceptual models used to  classify the research are simple in conception, but appropriate for use. The data base (originally enormous, then reduced as more evaluation criteria were introduced) is original, with potentially useful information. Some parts raise further questions. I list them below, which might help the authors in a rewrite:

The authors could also more clearly point out that the 165 studies are an underestimate, due to: (a) the limitations of the list of methods, (b) the limitations of the list of migration- and remittance-related concepts, (c) the limitation to the language EN, and (d) the limitations of the databases.

On p. 6 (search strategy), perhaps the authors should indicate whether the "advanced search" in the three databases uses the full text of the papers or just the metadata or the title…. ?

The distribution of identified studies by year should be commented in more detail.

This is an analysis with a (potential) policy vision at the beginning, but the policy-related presentation is not so solid and structured at its core. A section with some comments on key policy findings from the revised literature should be added. These findings relate to the fiscal and monetary spheres on the one hand, but also to the commercial sphere on the other. More isolated findings could be reported here, such as on financial innovation, sustainability strategies, or the policy agenda on labor, etc.

It is not so clear how the research methods are used in the literature. The paper should summarize and discuss observations on the scope and types of techniques used in the analysis of remittances and migration processes and outcomes. This could reveal a trend or shift in the techniques used in the selected papers.

The authors could also work to include the references of the 88 studies considered in the reference list.

The English language is proficient. Some minor modifications could be made to enhance the English language.

Author Response

(The authors gave the same response as above.)

Reviewer 3 Report

The authors attempted to study the impact of remittances on local economic development in poor mountainous areas using a literature review approach. This research method is novel and the research topic is engaging.

However, the authors' research methodology is simple and their conclusions are too broad. It did not add anything new to the existing body of scholarship. Nor did I gain new knowledge from it. The article is still a long way from the level of a thoroughly reputable academic journal. 

There is still a long way to go before it can be published.

Therefore, I decided to reject it.

Specific recommendations:

1, Focus on the scope of the study and analyze and study the content of the study in more depth;

2, Discuss the findings of the study more fully;

3, The article's value in the overall academic system should be fully explained and demonstrated.

4, The citation format of the article should be following the requirements of the journal to which the article is submitted.

Please edit and revise your article carefully and try to express yourself in the common language of academia.

Author Response

(The authors gave the same response as above.)

Reviewer 4 Report

The authors presented research on a topical issue that is of great importance for the economies of mountainous regions, which are faced with mass migration of the population. The article presents the impact of remittances on the living standards of the population of mountainous areas based on a review of the literature. The results of the study have important recommendations for scientists, financiers, and politicians. A bibliometric analysis of coworking showed the most studied areas such as gender, food security and migrant households. The article is recommended for publication.

Author Response

Thank you for the compliments.

Round 2

Reviewer 1 Report

Some improvements in the paper content were managed, even a paper with a topic so attractive as this, can be more improved.

A main recommendation is to arrange the paper in order to be easier for reading and understand, of course.

All figures can be better arranged and explained.

Author Response

Thank you. As suggested, we have constructed Excel graphs for Figures 4 and 5. However, we are unsure of what we have to do with the VOSViewer output. In the last submission, we were advised to include screenshots. We will be glad if you can shed more light on this. In the meantime, we have cropped the screenshots as in Figures 7 and 8.

Thank you
